# Effects of Miscut on Step Instabilities in Homo-Epitaxially Grown GaN

**DOI:** 10.3390/nano14090748

**Published:** 2024-04-25

**Authors:** Peng Wu, Jianping Liu, Fangzhi Li, Xiaoyu Ren, Aiqin Tian, Wei Zhou, Fan Zhang, Xuan Li, Bolin Zhou, Masao Ikeda, Hui Yang

**Affiliations:** 1Key Laboratory of Nanodevices and Applications, Suzhou Institute of Nano-Tech and Nano-Bionics, Chinese Academy of Sciences, Suzhou 215123, China; pwu2018@sinano.ac.cn (P.W.); fli2020@sinano.ac.cn (F.L.); xren2014@sinano.ac.cn (X.R.); aqtian2012@sinano.ac.cn (A.T.); wzhou2015@sinano.ac.cn (W.Z.); fzhang2019@sinano.ac.cn (F.Z.); xli2019@sinano.ac.cn (X.L.); bzhou2022@sinano.ac.cn (B.Z.); mikeda2013@sinano.ac.cn (M.I.); 2School of Physical Science and Technology, ShanghaiTech University, Shanghai 201210, China; 3Shanghai Advanced Research Institute, Chinese Academy of Sciences, Shanghai 201210, China; 4University of Chinese Academy of Sciences, Beijing 100049, China

**Keywords:** GaN, step bunching, step meandering, Ehrlich–Schwoebel (E-S) barrier, miscut, MOCVD, AFM

## Abstract

The rough morphology at the growth surface results in the non-uniform distribution of indium composition, intentionally or unintentionally doped impurity, and thus impacts the performance of GaN-based optoelectronic and vertical power electronic devices. We observed the morphologies of unintentionally doped GaN homo-epitaxially grown via MOCVD and identified the relations between rough surfaces and the miscut angle and direction of the substrate. The growth kinetics under the effect of the Ehrlich–Schwoebel barrier were studied, and it was found that asymmetric step motions in samples with a large miscut angle or those grown at high temperature were the causes of step-bunching. Meandering steps were believed to be caused by surface free energy minimization for steps with wide terraces or deviating from the $\left[1\bar{1}00\right]$ m-direction.

## 1. Introduction

Nowadays, GaN is receiving more and more attention due to its application prospects in electronic and optoelectronic devices. The application of high-quality freestanding GaN substrate is essential in further improving device performance for devices such as laser diodes and vertical power diodes [1,2,3,4,5,6,7]. However, numerous studies have revealed that device performance deteriorates due to the rough surfaces induced by meandering steps and macro-steps tens of monolayers high. Kizilyalli et al. found that the wavy morphology of homo-epitaxially grown GaN Shottky diodes and p-n diodes grown via MOCVD caused large current non-uniformities [8]. Hayashi et al. discovered that a striped morphology induced a current-crowding effect in vertical GaN p-n junction diodes [9]. GaN layers are often used as the basis for low-dimensional InGaN/GaN heterostructures. Indium composition fluctuations caused by a rough surface have been observed by many researchers, using measurements such as cathodoluminescence mapping [10,11,12,13,14,15] and scanning near-field optical microscopy [16]. According to Bales and Zangwill, wavy surfaces including meandering and bunching steps are mostly caused by morphological instabilities under the effect of adatom incorporation asymmetry at the steps [17]. Generally, the energy barrier for adatom crossing over a step is called the E-S barrier and can be classified into three types, as shown in Figure 1 [18,19,20]. Many results based on Monte Carlo simulations have verified that steps under an inverse E-S barrier easily bunch to form macro-steps [21,22,23,24,25]. However, the relations among the E-S barrier, miscut, and step instabilities are rarely studied with a focus on the aspect of growth kinetics. In this work, we found the relations between miscut and step instabilities in unintentionally doped GaN grown via MOCVD. The step kinetics affected by the E-S barrier were studied based on the step motion model proposed by Schwoebel and Shipsey [20,26]. The steps with wider terraces and those propagating toward directions deviating from the $\left[1\bar{1}00\right]$ m-axis were believed to be unstable.

## 2. Materials and Methods

The GaN substrate used in this paper has threading dislocation density of the about 10^6^ cm^−2^ and bow of 15 μm. The misorientation angle (also called miscut angle) and directions are nonuniform across the 2-inch substrate due to the crystalline bowing during substrate epitaxy, and hence the miscut angle varies from 0.22° to 0.55°. Therefore, the effect of miscuts on homo-epi GaN morphology can be studied on such a GaN substrate with quite different miscuts in different areas. Unintentionally doped GaN (uGaN) films with a thickness of about 100 nm were grown twice, subsequently, at different growth temperatures using a metalorganic chemical vapor deposition (MOCVD) reactor. Trimethylgallium (TMGa) and ammonia were used as source gases, while hydrogen was used as carrier gas. The local miscut information was obtained via the analysis of XRD rocking curves by rotating the substrate around surface normal. Surface morphologies were studied using an atomic force microscope (AFM, Bruker Dimension ICON, Billerica, MA, USA) in tapping mode. AFM amplitude images were used to clearly show atomic step features even on a surface tens of microns large, with large height differences.

## 3. Results

AFM images of samples grown at different temperature in the areas with a different miscut angle toward the $\left[1\bar{1}00\right]$ m-direction are shown in Figure 2. At 888 °C, the surface morphology of uGaN in the area with a miscut angle of 0.22° features meandering steps, as shown in Figure 2a, which consist of steps perpendicular to the $\left[1\bar{1}00\right]$ m-direction with a terrace width of about 45 nm and steps perpendicular to the $\left[10\bar{1}0\right]$ m-direction with a terrace width of about 76 nm. The spacing between interlacing steps seems to be random. For the areas with a larger miscut angle, the surface morphologies show uniform and straight atomic steps, as shown in Figure 2b,c. The average terrace widths in both areas agree well with the miscut angle. For samples grown at 1040 °C, V-pits induced by threading dislocations disappear. The surface in the area with a miscut angle of 0.34° maintains a step-flow morphology and has a decrease in roughness due to the elimination of V-pits. The surface in the area with a miscut angle of 0.22° is smoother due to the suppression of the step-meandering phenomenon, as shown in Figure 2d. In contrast, step-bunching with a maximum step height of about 9.7 nm takes place in the area with a miscut angle of 0.55°, as shown in Figure 2f. This is a very common phenomenon for surfaces with narrower terraces due to the increased miscut angle.

Figure 3 shows the morphology in the area misoriented toward the direction deviating from the $\left[1\bar{1}00\right]$ m-direction. The wave-shaped steps in Figure 3a also suggests a slight step-meandering phenomenon with quite a uniform spacing of interlacing steps, of about 2 μm. For the area with a larger miscut angle (0.43°) misoriented toward the direction deviating 36° from the $\left[1\bar{1}00\right]$ m-direction, the step-flow morphology with uniform and straight terraces is shown in Figure 3b. At the growth temperature of 1040 °C, both areas with miscut angles of 0.35° and 0.43° show a step-bunching morphology. The maximum height of bunched steps is 5 nm for the area with a miscut angle of 0.35° and 5.2 nm for the area with a miscut angle of 0.43°.

By combining our analysis of Figure 2 and Figure 3, we can see that areas with a miscut angle larger than 0.34° all show a step-bunching morphology when grown at 1040 °C. The larger the miscut angle, the rougher the surface according to the RMS values. Moreover, the height differences of the bunched steps are bigger in the area with a larger miscut angle. Hence, the morphological instability induced by step bunching is more severe at a larger miscut angle. The morphological difference in Figure 2e and Figure 3c suggests that meandering steps are more easily bunched than straight steps under the same growth condition. Comparing all AFM images of uGaN grown at 888 °C, we can conclude that the step-meandering phenomenon easily occurs at the surfaces of steps with wider terraces or deviating from the $\left[1\bar{1}00\right]$ m-direction and can be effectively suppressed by increasing the growth temperature.

## 4. Discussion

To explain the relations between the step-bunching morphology and the miscut angle, we applied the step motion model introduced by Schwoebel and Shipsey to study step kinetics under the effect of the E-S barrier [20]. The detailed simulated parameters can be found in our previous paper [27]. As shown in Figure 4, only a small portion of adatoms reaching the step can be incorporated into the step due to behaviors such as reflection from the step, transmission across the step, and diffusion along the step [28]. Hence, the incorporation probability of adatoms from the lower terrace is defined as k_+_, while that from the upper terrace is defined as k_−_. Under a positive E-S barrier, the adatoms from the upper terrace have more difficulty being incorporated at the step, which makes k_+_ larger than k_−_. While the E-S barrier occurs at the lower terrace (inverse), k_+_ is smaller than k_−_. There is no E-S barrier when k_+_ equals k_−_. When the widths of all terraces are equal at the initiate state, all steps advance at the same speed and remain unaffected by the E-S barrier and adatom diffusion length. However, the above ideal situation does not occur, and the terraces of steps fluctuate around the average value on the real surfaces. Therefore, a little deviation is applied to one of the steps to make it travel 1/4 of the average terrace width in the initial state.

Figure 5 shows the simulated results under the circumstances of there being no E-S barrier with a different adatom diffusion length. When the adatom diffusion length is insufficient in the red line, the effect of the intentionally introduced deviation is concealed because the steps maintain the same speed. The terrace around the deviated step tends to broaden or narrow in the blue line when the adatom diffusion length is equal to the average terrace width. Step bunching occurs with a much larger adatom diffusion length, as shown via the green line. Therefore, the effects of the E-S barrier and the deviation in the terrace width are more significant with a larger adatom diffusion length.

In the following simulations, the adatom diffusion length is set to be double that of the average terrace width to clearly observe the effect of the E-S barrier. The higher the E-S barrier height, the greater the difference in adatom incorporation probability from the lower and upper terrace. Hence, we can adjust the value of k_+_ and k_−_ to study the effect of different E-S barrier heights. As shown in Figure 6, step bunching occurs under the circumstance of a free E-S barrier (green line), while it disappears under the circumstance of a relatively low positive E-S barrier (blue line). When it comes to the extreme case that no adatoms at the upper terrace can overcome the E-S barrier (red line), the terrace widths of all steps become equal. This obviously shows that a relatively large positive E-S barrier is helpful in suppressing fluctuations in the terrace width and creates a stable surface morphology. In contrast, steps easily bunch with an inverse E-S barrier, as shown in Figure 7. The maximal layers of bunched steps rapidly increase with the height of the inverse E-S barrier (5 for the blue line and 8 for the green line).

According to many studies, both the adatom diffusion length *λ* and the E-S barrier are closely related to growth conditions, such as the temperature [29,30,31]. According to Einstein’s relation [32],
(1)$$\lambda =\sqrt{D\tau }=\lambda_{0}\exp\;\frac{E_{a}-E_{d}}{2k_{B}T}$$
where *D* is the diffusion coefficient, *τ* is the mean time of surface diffusion, *λ*_0_ is a merged effective elementary jump distance, *k_B_* is the Boltzmann constant, *T* is the growth temperature, *E_a_* is the absorption energy and equal to the barrier to be surmounted for desorption, and *E_d_* is the energy barrier for diffusion. In MOCVD, GaN epitaxy is maintained in a diffusion-limited regime, which is grown at a low temperature, much lower than the temperature for evaporation. Therefore, *E_a_* is usually much larger than *E_d_* and also several times larger than the thermal energies *k_B_T*. The desorption of adatoms becomes negligible, and the residence time is large. Hence, *λ* is proportional to −*E_d_*/(2*k_B_T*) and, consequently, the adatom diffusion length is positively related to the growth temperature. The positive E-S barrier can be lowered by an increased temperature and can lead to a growth mode transition from island to step-flow. Hence, the elevated growth temperature increases the adatom diffusion length and lowers the positive E-S barrier or even makes it inverse, resulting in more adatoms from the upper terrace being incorporated into the steps and forming a step-bunching morphology [33].

Regarding the step-meandering phenomenon, the one-dimensional step motion model mentioned above is not applicable, and hence it is discussed qualitatively. In general, meandered steps can be caused by extrinsic perturbations at the surface, such as defects, impurities, or local fluctuation, or by intrinsic perturbations, such as the E-S barrier and surface tension [11]. The extrinsic perturbations are irregular and randomly distributed, and thus we focus on the influence of the E-S barrier and surface tension. Numerous theoretical studies have shown that the existence of the E-S barrier aggravates the fluctuation of the step shape [17,34,35,36,37,38,39]. According to Jeong and Williams, the surface free energy *γ_α_* can be described by the free energy of the terrace *γ*_0_, the free energy of the step *β*, and the step–step interaction g as a function of the miscut angle α in the following way [40]:(2)$$\gamma_{\alpha }=\gamma_{0}+\frac{\beta }{h}\left|\tan\;\alpha \right|+g{\left|\tan\;\alpha \right|}^{3}$$
where *h* represents the atomic step height of a half-unit cell. For a misoriented surface, it can adjust the dangling bonds on the terraces and at the steps or modulate the kink distributions along the steps to make it energetically favorable in specific growth conditions. However, this kind of change may be insufficient sometimes, such as for surfaces covered by ultra-wide terraces or steps with a large amount of free energy. In this case, the surface will break up into facets of the neighboring orientations that are represented in the equilibrium crystal shape [41]. According to the crystal structure of GaN, steps toward the $\left[1\bar{1}00\right]$ m-direction and its equivalent directions have the lowest energy [42]. Therefore, the free energy of steps deviated from the $\left[1\bar{1}00\right]$ m-direction *β*’ is much larger than that toward the $\left[1\bar{1}00\right]$ m-direction *β*. On the other hand, the free energy of terraces is proportional to their width. Hence, the free energy of wide terraces *γ*_0′_ is much larger than that of narrow terraces *γ*_0_. Therefore, steps deviated from the $\left[1\bar{1}00\right]$ m-direction or with wide terraces are unstable and readily swerve into neighboring orientations or form meandering edges.

## 5. Conclusions

In summary, the surface morphologies of unintentionally doped GaN are observed in the freestanding substrate with different miscut angles and miscut directions. Step instabilities including step meandering and step bunching are found to be associated with the miscut angle, miscut direction, and growth temperature. Bunched steps are easily generated in samples with a large miscut angle or those grown at a high temperature. According to the simulated results, increasing the growth temperature results in a longer adatom diffusion length, accompanied by changes in the E-S barrier, which makes it easy for step bunching to occur. Steps with wide terraces or those deviating from $\left[1\bar{1}00\right]$ m-direction are believed to have a higher free energy and, hence, easily meander. These findings indicate that a substrate miscut angle and direction and growth conditions are essential in obtaining a smooth surface.

## Figures and Tables

**Figure 1 nanomaterials-14-00748-f001:**
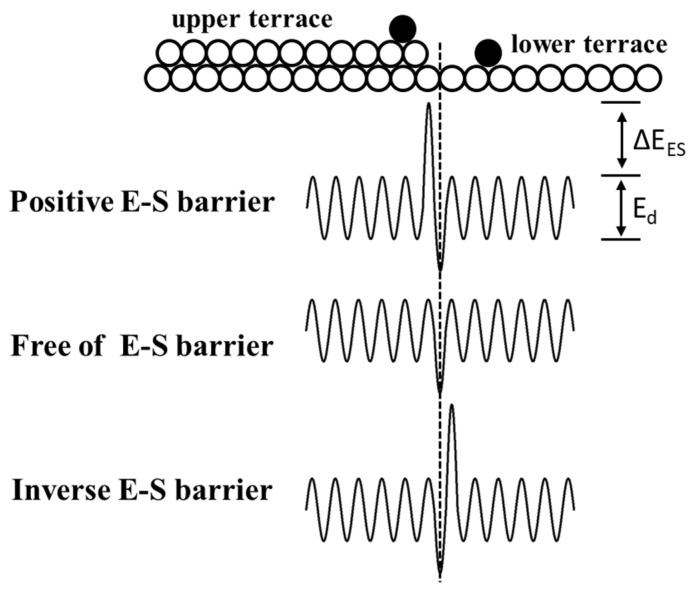
The energy barrier ΔE_ES_ for adatoms at the upper terrace (positive) or at the lower terrace (inverse) to overcome when incorporated into the step. There is no E-S barrier when the incorporation possibility for adatoms from both sides of the step is equal. E_d_ is the diffusion barrier energy.

**Figure 2 nanomaterials-14-00748-f002:**
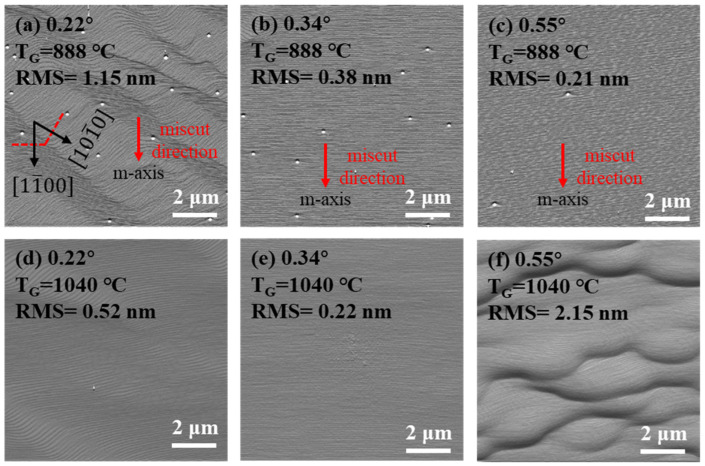
This figure shows 10 × 10 μm^2^ AFM amplitude images of uGaN surfaces at the areas with different miscut angles toward the $\left[1\bar{1}00\right]$ m-direction. The first line of the images is taken after first growth at a growth temperature of 888 °C, while the second line is taken after second growth at 1040 °C. (**a**,**d**) 0.22°; (**b**,**e**) 0.34°; (**c**,**f**) 0.55°.

**Figure 3 nanomaterials-14-00748-f003:**
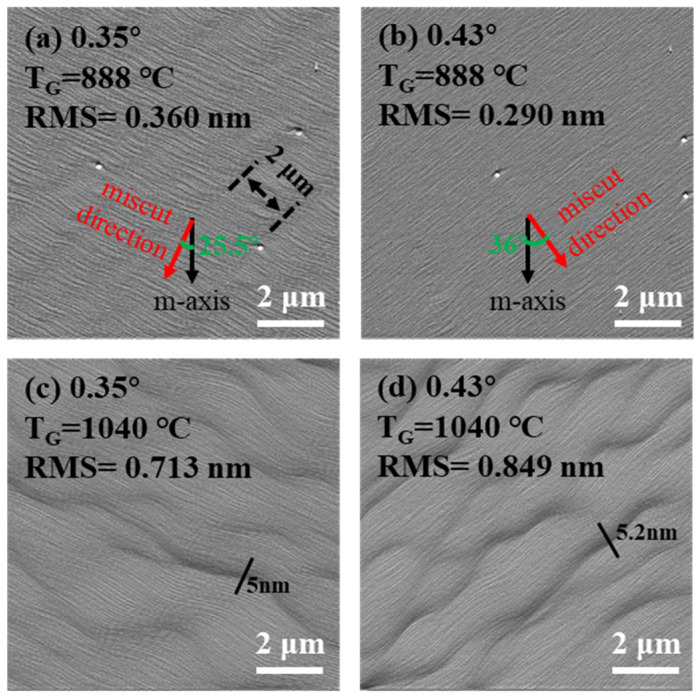
This figure shows 10 × 10 μm^2^ AFM amplitude images of uGaN surfaces in the areas with different miscut angles deviated from the $\left[1\bar{1}00\right]$ m-direction. The first line of the images is taken after first growth at a growth temperature of 888 °C, while the second line is taken after second growth at 1040 °C. (**a**,**c**) 0.34° toward (−a + m)-direction; (**b**,**d**) 0.43° toward (a + m)-direction.

**Figure 4 nanomaterials-14-00748-f004:**
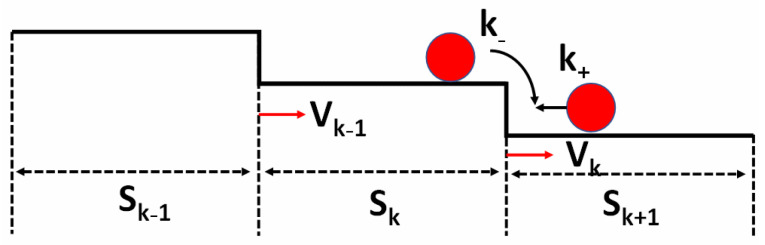
Step motion model on a vicinal surface. v_k_ is the velocity of the kth step, while the width of its upper and lower terrace is s_k_ and s_k−1_, respectively. k_+_ and k_−_ represent the adatom incorporation probability from the lower and upper terraces, respectively. v_k+1_ and s_k+1_ represent the velocity of the (k + 1)th step and the width of its lower terrace.

**Figure 5 nanomaterials-14-00748-f005:**
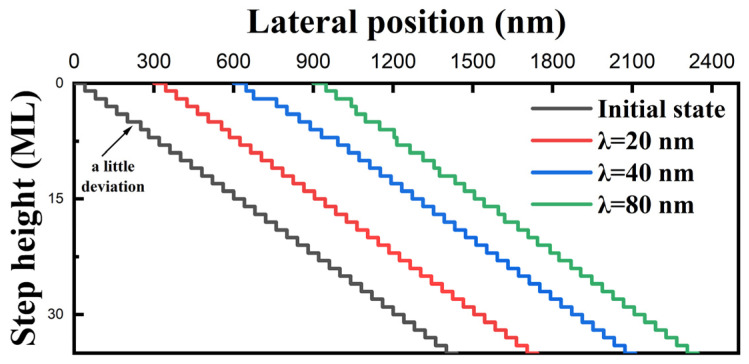
Simulated results with a different adatom diffusion length when there is no E-S barrier (k_+_ = k_−_ = 0.1).

**Figure 6 nanomaterials-14-00748-f006:**
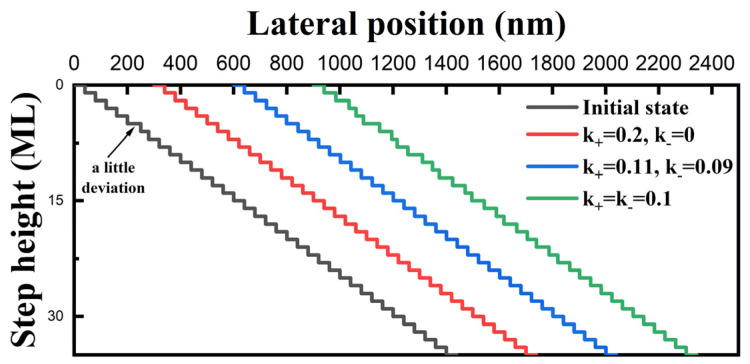
Simulated results showing the effect of a positive E-S barrier on step motions (λ = 80 nm). The green line is under the circumstance that there is no E-S barrier, while the red line is under an extreme case when the energy barrier at the upper terrace is so large that no adatoms can cross over and be incorporated into the step. A relatively small E-S barrier is set for the blue line.

**Figure 7 nanomaterials-14-00748-f007:**
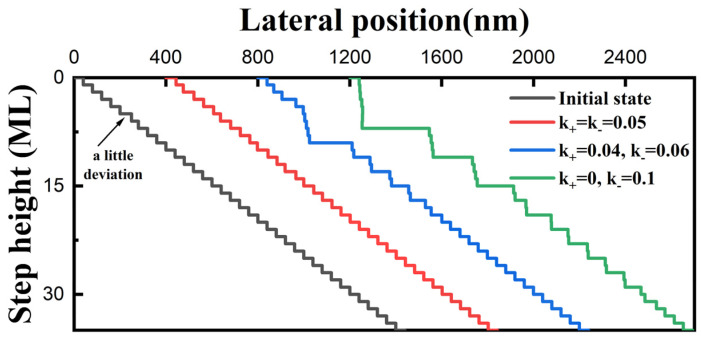
Simulated results showing the effects of an inverse E-S barrier on step motions (λ = 80 nm). The red line is under the circumstance of there being no E-S barrier, while the green line is in an extreme case where the energy barrier at the lower terrace is so large that no adatoms can cross over and be incorporated into the step. A relatively small E-S barrier is set for the blue line.

## Data Availability

The data presented in this study are available on request from the corresponding author.

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
