# Peer review of "Effects of Miscut on Step Instabilities in Homo-Epitaxially Grown GaN"

_nanomaterials, 2024, doi:10.3390/nano14090748_

Round 1

Reviewer 1 Report

Comments and Suggestions for Authors

This article demonstrated that the morphologies of unintentionally doped GaN homo-epitaxially grown by MOCVD and find the relations between rough surfaces and the miscut angle and direction of the substrate. The current paper can be certainly published with little corrections. The authors might refine the places that needed further elaboration. Sufficient explanation helps forming a more precise and convincing paper.

It lacks detailed scale and unit of the lateral position axes from Fig. 5 to 7.

It is suggested to clarify the parameters of ΔEES (Ehrlich-Schwoebel barrier) and Ediff (diffuse barrier) on the terrace in the simulated model in Fig. 1. The simulation tools should be mentioned in this step-flow growth techniques.

The authors found that asymmetric step motions on samples with large miscut angle or grown at high temperature are the causes of step-bunching. The results should be specific. What is the influence of growth temperature other than 888 and 1040 oC?

Is there any transition growth temperature effects on step-bunching? Elaborated clarification is required.

For a better introduction and clearer understanding of the methodology, the following references can be used:

• Yamada, Hisashi, et al. "Impact of substrate miscut on the characteristic of m-plane InGaN/GaN light emitting diodes." Japanese Journal of Applied Physics 46.12L (2007): L1117.

• Lin, You-Da, et al. "Blue-green InGaN/GaN laser diodes on miscut m-plane GaN substrate." Applied physics express 2.8 (2009): 082102. •

Yu-Sheng Huang, et al. ”Nanostructure analysis of InGaN/GaN QWs based on semi-polar face GaN nanorods,” Optical Materials Express Vol. 7, No. 2 pp. 320-328 (2017).

Comments on the Quality of English Language

This article demonstrated that the morphologies of unintentionally doped GaN homo-epitaxially grown by MOCVD and find the relations between rough surfaces and the miscut angle and direction of the substrate. The current paper can be certainly published with little corrections. The authors might refine the places that needed further elaboration. Sufficient explanation helps forming a more precise and convincing paper.

It lacks detailed scale and unit of the lateral position axes from Fig. 5 to 7.

It is suggested to clarify the parameters of ΔEES (Ehrlich-Schwoebel barrier) and Ediff (diffuse barrier) on the terrace in the simulated model in Fig. 1. The simulation tools should be mentioned in this step-flow growth techniques.

The authors found that asymmetric step motions on samples with large miscut angle or grown at high temperature are the causes of step-bunching. The results should be specific. What is the influence of growth temperature other than 888 and 1040 oC? Is there any transition growth temperature effects on step-bunching? Elaborated clarification is required.

For a better introduction and clearer understanding of the methodology, the following references can be used:

·        Zhi Ting Ye, et al. Nanoparticle-Doped Polydimethylsiloxane Fluid Enhances the Optical Performance of AlGaN-Based Deep-Ultraviolet Light-Emitting Diodes,” Nanoscale Research Letters 14, 236 (2019)

·        Shih-Wei Feng et al. “Precursor Duration and Thermal Annealing Effects in InGaN/GaN Multiple Quantum Wells grown on nitrogen-polar GaN templates by a Pulsed Metalorganic Chemical Vapor Deposition,” ECS Journal of Solid State Science and Technology 7(10), R161-R165 (2018).

·        Yu-Sheng Huang, et al. ”Nanostructure analysis of InGaN/GaN QWs based on semi-polar face GaN nanorods,” Optical Materials Express Vol. 7, No. 2 pp. 320-328 (2017).

Author Response

Point 1, 2 and 5:

It lacks detailed scale and unit of the lateral position axes from Fig. 5 to 7.

It is suggested to clarify the parameters of ΔEES (Ehrlich-Schwoebel barrier) and Ediff (diffuse barrier) on the terrace in the simulated model in Fig. 1. The simulation tools should be mentioned in this step-flow growth techniques.

For a better introduction and clearer understanding of the methodology, the following references can be used.

Response : Thank you for your kind and perspective advices on Figs. 1, 5, 6 and 7 and references. I have modified the figures and added the recommendated references in the manuscript. The simulation is simply based on step motion model proposed by Schwoebel and Shipsey according to their paper “Step motion on crystal surfaces”. You can find detailed simulated parameters in our previous paper as shown in Ref. 25.

Point 3: The authors found that asymmetric step motions on samples with large miscut angle or grown at high temperature are the causes of step-bunching. The results should be specific. What is the influence of growth temperature other than 888 and 1040 ℃?

Response: Thank you for your comments.  Because bulk GaN substrate is very expensive, we only studied two diffferent growth temperatures at this point. We will study the influence of growth temperature other than 888 and 1040 ℃ in the future.

Point 4: Is there any transition growth temperature effects on step-bunching? Elaborated clarification is required.

Response: Thank you for your comments. According to our simulated results, elevated growth temperature will not only increase the adatom diffussion length and hence make the effects of E-S barrier more significant, but also change the barrier height difference for adatoms when ascending or descending steps. The relations of E-S barrier height and growth temperature can also be found in the thesis of Nils Asmus Kristian Kaufmann (https://doi.org/10.5075/epfl-thesis-5776).

Thank you very nuch for your review and comments. We hope the revised manuscript could be accepeted for publication. If not, we are glad to receive any further comments and advices so that we can improve our manuscript further.

Reviewer 2 Report

Comments and Suggestions for Authors

The work is devoted to studying the surface morphology of GaN grown by MOCVD. The relevance and demand of the work is beyond doubt. The work presents interesting experimental data showing a significant dependence of surface morphology on the misorientation angle, misorientation direction, and growth temperature. However, there are a number of significant comments to the interpretation given by the authors. Nevertheless, the paper deserves publication, mainly due to the extremely interesting experimental data.
Notes and recommendations:
1.    Just recommendation. I think that it will be good for paper visibility to use keywords like AFM and MOCVD.
2.    It seems a little strange to mention the heterogeneity of the In composition (both in the introduction and in the annotation) without an explicit connection with InGaN structures. I would recommend pointing out that GaN layers are often used as the basis for low-dimensional InGaN/GaN heterostructures for thick InGaN layers.
3.    Lines 43 and 58. Typo in the word “unintentionally”.
4.    What is the reason for the change in the misorientation angle of the substrate? Was this intentionally put in during the cut? In the case of GaN/sapphire epitaxial layers, the curvature of the substrate could be explained by residual thermal stresses, which, of course, does not apply to a matched GaN substrate.
5.    What polarity of the substrate was used? Will the behavior of the relief during growth depend on the polarity of the material?
6.    From Figure 2(a), it is clear that those steps that the authors designate as perpendicular to the [10-10] direction actually go at a noticeable angle to this direction (red dotted line next to the [10-10] arrow). Perhaps I misunderstood the authors?
7.    The authors’ conclusion that the curvature of steps can be suppressed by increasing the growth temperature (lines 109-112) seems to me to be inconsistent with the AFM image data. As can be seen from Figures 2(f) and 3(c, d), an increase in temperature only leads to the development of relief.
8.    It is not entirely clear why the authors call the energy required for desorption as an absorption energy (Ea). Lines 174-175. In addition, it is not very clear what kind of desorption the authors are talking about. Do you mean the transition from the adatom state to vacuum? Or from the state embedded in a crystal to the state of a mobile adatom?
9.    The formulas are not numbered, which makes it difficult to read.
10.     The authors write: “Therefore, Ea is usually much larger than Ed and consequently adatom diffusion length is positively related to growth temperature.” however, from expression (1?) it follows that just the opposite - an increase in temperature leads to a decrease in the diffusion length, provided that Ea>>Ed. A detailed explanation is required.

Author Response

Point 1: Just recommendation. I think that it will be good for paper visibility to use keywords like AFM and MOCVD.

Point 3: Lines 43 and 58. Typo in the word “unintentionally”.

Point 9: The formulas are not numbered, which makes it difficult to read.

Response: Thank you very much for your review and for your comments. We have revised the manuscript according to your suggestions.

Point 2: It seems a little strange to mention the heterogeneity of the In composition (both in the introduction and in the annotation) without an explicit connection with InGaN structures. I would recommend pointing out that GaN layers are often used as the basis for low-dimensional InGaN/GaN heterostructures for thick InGaN layers.

Response: Thank you very much for your comments. We have added it in the revised manuscript.

Point 4: What is the reason for the change in the misorientation angle of the substrate? Was this intentionally put in during the cut? In the case of GaN/sapphire epitaxial layers, the curvature of the substrate could be explained by residual thermal stresses, which, of course, does not apply to a matched GaN substrate.

Response: The GaN substrate used in our experiments was fabricated on sapphire substrate by HVPE method. The misoriented GaN substrated is obtained after surface cleaning and chemical machanical polishing toward specific direction (  for the substrate used in this work). However, the crystallographic orientations of GaN surface is not congruent at different positions (which is called bow/curvature) due to hetero-epitaxially growth, which results into non-uniformly distributed miscut.

Point 5: What polarity of the substrate was used? Will the behavior of the relief during growth depend on the polarity of the material?

Response: The substrate used in this work is Ga-polar GaN. Due to different surface energy, adatoms is observed to have lower mobility and easily form hillock or finger-like morphology  for N-polar GaN according to many studies (10.1063/1.3006132, 10.1063/1.4818322 and 10.1016/j.apsusc.2019.04.082). Hence, things is pretty different when it comes to N-polar GaN and it surely needs deep research.

Point 6: From Figure 2(a), it is clear that those steps that the authors designate as perpendicular to the [10-10] direction actually go at a noticeable angle to this direction (red dotted line next to the [10-10] arrow). Perhaps I misunderstood the authors?

Response: The steps indeed seems to propagate at a slight angle to the labelled  direction. But we regard it as deviations because those steps is not at the same plane with the steps propagating toward  direction. According to crystal structure of wurtzite GaN,  and  are equivalent crystallographic directions. Hence, steps meander from  direction to  direction according to minimum energy principle.

Point 7: The authors’ conclusion that the curvature of steps can be suppressed by increasing the growth temperature (lines 109-112) seems to me to be inconsistent with the AFM image data. As can be seen from Figures 2(f) and 3(c, d), an increase in temperature only leads to the development of relief.

Response: We can easily ditinguish that the curved steps in Figure 2(a) are different from that in Figures 2(f), 3(c) and 3(d). It is stair-like and has specific direction for the steps in Figure 2(a) (which is called step meandering) and is wandering for the steps in Figures 2(f), 3(c) and 3(d) (which is called step bunching). Our conclusion is that increasing the growth temperature can help suppress step meandering and promote step bunching. Another conclusion is that steps propagating deviated from  m-direction are easier bunched that that propagating to m-direction.

Point 8: It is not entirely clear why the authors call the energy required for desorption as an absorption energy (Ea). Lines 174-175. In addition, it is not very clear what kind of desorption the authors are talking about. Do you mean the transition from the adatom state to vacuum? Or from the state embedded in a crystal to the state of a mobile adatom?

Response: As mentioned in Ref. 28, desorption energy is the energetic barrier to be surmounted for adatoms to escape from the surface to the gas phase. This kind of energy is equal to adsorption energy which is the energy acquired for the transition from the adatoms state to gas phase.

Point 10: The authors write: “Therefore, Ea is usually much larger than Ed and consequently adatom diffusion length is positively related to growth temperature.” however, from expression (1?) it follows that just the opposite - an increase in temperature leads to a decrease in the diffusion length, provided that Ea>>Ed. A detailed explanation is required.

Response: Thanks for pointing out this mistake. We have corrected it and added Ref. 30 where it originates from.

Thank you very nuch for your review and comments. We hope the revised manuscript could be accepeted for publication. If not, we are glad to receive any further comments and advices so that we can improve our manuscript further.

Round 2

Reviewer 1 Report

Comments and Suggestions for Authors

Accept in present form.

Author Response

I haven't seen any advices from Reviewer 1 at round 2 review process. If you have any questions please don't hesitate to ask.

Reviewer 2 Report

Comments and Suggestions for Authors

Authors are satisfactory answer on most comments, but, unfortunately, some of them still not fixed. In particular:

Point 10: The authors write: “Therefore, Ea is usually much larger than Ed and consequently adatom diffusion length is positively related to growth temperature.” however, from expression (1?) it follows that just the opposite - an increase in temperature leads to a decrease in the diffusion length, provided that Ea>>Ed. A detailed explanation is required.

Response: Thanks for pointing out this mistake. We have corrected it and added Ref. 30 where it originates from.

However, authors doesn't change anything in the text. The problem is not fixed. Also, it is true for Point 9 about formulas numbering.

Please, check it carefully.

Round 3

Reviewer 2 Report

Comments and Suggestions for Authors

Authors takes good response for all comments. Paper can be published.